# Dual and Opposite Costimulatory Targeting with a Novel Human Fusion Recombinant Protein Effectively Prevents Renal Warm Ischemia Reperfusion Injury and Allograft Rejection in Murine Models

**DOI:** 10.3390/ijms22031216

**Published:** 2021-01-26

**Authors:** Jordi Guiteras, Laura De Ramon, Elena Crespo, Nuria Bolaños, Silvia Barcelo-Batllori, Laura Martinez-Valenzuela, Pere Fontova, Marta Jarque, Alba Torija, Oriol Bestard, David Resina, Josep M Grinyó, Joan Torras

**Affiliations:** 1Experimental Nephrology Laboratory, Institut d’Investigació Biomèdica de Bellvitge (IDIBELL), L’Hospitalet de Llobregat, 08907 Barcelona, Spain; jguiteras@idibell.cat (J.G.); deramon.laura@gmail.com (L.D.R.); ecrespo@idibell.cat (E.C.); nbolanos@idibell.cat (N.B.); lauramartinezval252@gmail.com (L.M.-V.); pfontova@idibell.cat (P.F.); mjarque@idibell.cat (M.J.); atorija@idibell.cat (A.T.); 2Fundació Bosch i Gimpera, University of Barcelona, 08028 Barcelona, Spain; 3Molecular Interactions Unit, Institut d’Investigació Biomèdica de Bellvitge (IDIBELL) Scientific-Technical Services, L’Hospitalet de Llobregat, 08907 Barcelona, Spain; sbarcelo@idibell.cat; 4Nephrology Department, Bellvitge University Hospital, L’Hospitalet de Llobregat, 08907 Barcelona, Spain; obestard@bellvitgehospital.cat; 5Faculty of Medicine, Bellvitge Campus, University of Barcelona, L’Hospitalet de Llobregat, 08907 Barcelona, Spain; 6Bioingenium S.L., Barcelona Science Park, 08028 Barcelona, Spain; dresina@bioingenium.net

**Keywords:** costimulation, coinhibition, ischemia-reperfusion injury, kidney transplant, SPR, protein binding affinity, innate immunity, adaptive immunity, inflammation

## Abstract

Many studies have shown both the CD28—D80/86 costimulatory pathway and the PD-1—PD-L1/L2 coinhibitory pathway to be important signals in modulating or decreasing the inflammatory profile in ischemia-reperfusion injury (IRI) or in a solid organ transplant setting. The importance of these two opposing pathways and their potential synergistic effect led our group to design a human fusion recombinant protein with CTLA4 and PD-L2 domains named HYBRI. The objective of our study was to determine the HYBRI binding to the postulated ligands of CTLA4 (CD80) and PD-L2 (PD-1) using the Surface Plasmon Resonance technique and to evaluate the in vivo HYBRI effects on two representative kidney inflammatory models—rat renal IRI and allogeneic kidney transplant. The Surface Plasmon Resonance assay demonstrated the avidity and binding of HYBRI to its targets. HYBRI treatment in the models exerted a high functional and morphological improvement. HYBRI produced a significant amelioration of renal function on day one and two after bilateral warm ischemia and on days seven and nine after transplant, clearly prolonging the animal survival in a life-sustaining renal allograft model. In both models, a significant reduction in histological damage and CD3 and CD68 infiltrating cells was observed. HYBRI decreased the circulating inflammatory cytokines and enriched the FoxP3 peripheral circulating, apart from reducing renal inflammation. In conclusion, the dual and opposite costimulatory targeting with that novel protein offers a good microenvironment profile to protect the ischemic process in the kidney and to prevent the kidney rejection, increasing the animal’s chances of survival. HYBRI largely prevents the progression of inflammation in these rat models.

## 1. Introduction

Delayed graft function (DGF) is the clinical manifestation of ischemia-reperfusion injury (IRI) in human renal transplantation. The clinical impact of DGF on transplant-related outcomes is an increase in the risk of rejection, inferior allograft function, difficult management of immunosuppression in the early post-transplant phase, and the need for repeated renal biopsies with their related associated complications [1,2,3].

IRI causes a series of humoral and cellular responses, producing complex and diverse pathophysiological pathways of damage to the organ [4,5]. The occlusion of the arterial blood supply leads to tissue hypoxia and the imbalance of metabolic demand, thus activating cell death programs, endothelial dysfunction, transcriptional reprogramming, the expression of cytokines, leukocyte-endothelial adhesion, macrophage and lymphocyte activation, and the generation of small amounts of reactive oxygen species (ROS) [6,7,8,9]. Additionally, the restoration of blood flow and reoxygenation is frequently associated with an exacerbation of tissue injury and intense inflammatory response [10]. During reperfusion, the restoration of oxygen levels further damages cells exposed to previous ischemia, leading to the production of large amounts of ROS that contribute to cell membrane and DNA damage. Reperfusion damage results in an autoimmune response, which includes mostly natural antibody recognition and complement cascade activation [11]. This induces cell death and the activation of the innate and, afterwards, the adaptive immune system [6]. Regarding adaptive immunity, multiple roles of T cells have been described in studies of ischemia, showing evidences in both antigen-specific and antigen-independent mechanisms of activation [1,12,13].

Given the crucial role that costimulatory signals play in T cell-mediated immune responses, several costimulatory pathways have been targeted using monoclonal antibodies and fusion proteins to induce immunosuppression. Currently, it is well known that the so-called costimulatory pathways constitute a complex set of stimulatory and inhibitory signals which together fine-tune cell responses. To induce immunosuppression, several of these targets have been directed [14,15,16,17]. Some studies reported that CTLA4 blocks the costimulatory CD28 pathway by binding to the ligands CD80 and CD86. This blockade prevents the clonal expansion of effector T cells, leading them to an anergy or apoptotic state [18,19,20]. The costimulatory blocker Belatacept (a mutated version of CTLA4Ig) was approved for immunosuppression in renal transplantation in the early 2000s. The observed advantages of Belatacept over cyclosporine include better graft function, the preservation of renal structure, and improved cardiovascular risk profile [21]. Concerns associated with Belatacept are a higher frequency of cellular rejection episodes, but protocol biopsies at 1 year showed a lower incidence of chronic allograft nephropathy with Belatacept compared to cyclosporine [22]. According to Parsons et al., Belatacept can also reduce Human leukocyte antigen (HLA) class I antibodies in a significant proportion of highly sensitized recipients [23].

Recent studies have shown the immunosuppressive effects of exacerbating the coinhibitory pathway formed by PD-1 and the ligands PD-L1 and PD-L2 [24,25,26,27,28]. Early PD-1 expression has been demonstrated in renal allograft, which was considered essential to modulate T cell expansion and cytokine production. Blocking the PD-1/PD-L1 coinhibitory pathway during the first week after transplantation doubled the number of PD-1–expressing CD8 and CD4 cells infiltrating the graft, mainly in the interstitium [29]. Contrasting to PD-L1, PD-L2 is mainly expressed in the Antigen presenting cells (APC), does not link to CD80, and seems to be more significant in terms of protection against an ischemic or allogeneic insult [24,30,31,32]. Moreover, this pathway has been recently related to the regulatory T cell induction setting, a cellular subset implicated in the immune and inflammatory response regulation [33,34,35].

According to these T cell-modulating pathways, the concurrent blockade of the CD28—CD80/86 costimulatory pathway with CTLA4 and the stimulation of the PD-1 coinhibitory pathway with PD-L2 in the cell synapses could exert better immunosuppressive or anti-inflammatory effects [24,36]. In this regard, our group designed a novel human fusion protein construct to target these dual and opposite costimulatory signals. The production of the molecule consists of gen synthesis, cloning, and transient transfection with a stable Chinese hamster ovary (CHO) mammalian cell line. This recombinant protein design consists of an IgG1 FC linked to a CTLA4 molecule which is also bound to two PD-L2 molecules by polypeptide links (Figure 1).

This dual targeting may promote T cell anergy and apoptosis, enhance T cell exhaustion and T-reg expansion, and downregulate the CD80/86 expression in antigen-presenting cells [37,38,39]. This might suggest that lower concentrations of CTLA4 would be needed to inhibit the CD28—CD80/86 costimulatory synapsis. This potential dual targeting may close a synergistic circle to exert potent anti-inflammatory and immunosuppressive effects. We studied two experimental settings, the inflammatory environment of renal warm IRI and the renal allogeneic transplantation with cold ischemia, to evaluate the effect of our HYBRI protein.

## 2. Results

### 2.1. Analysis of HYBRI Binding Affinity to Murine CD80 and PD-1 Proteins Using the Surface Plasmon Resonance (SPR) Technique

SPR analysis was used to characterize the affinity and kinetics between the HYBRI construct and PD-1 and CD80. HYBRI was covalently bound to the CM5 sensor surface and the analytes (CD80 and PD-1) were injected in solution over the surface at concentrations ranging from 15.6 to 2000 nM. Sensorgrams (Figure 2) show a dose-dependent direct interaction between the analytes and HYBRI. Affinity constants determined in the steady state reflected a high affinity for both CD80 and PD1 binding to HYBRI (Table 1). Kinetics analysis revealed rapid association (Ka) and dissociation (Kd) for both CD80 and PD1. From these kinetic constants, affinity constants could be calculated (KD) and were in good agreement with the affinity constants determined from the steady-state analysis. Experiments were also conducted to assess the effect of the presence of PD-1 on CD80 binding to HYBRI, and there was found to be a lack of competition between the two molecules for HYBRI [40].

### 2.2. Bilateral Warm Ischemia Model in Native Rat Kidneys

#### 2.2.1. Effects of HYBRI on Renal Function

HYBRI therapy produced a significant improvement in kidney function the following three days after warm ischemia, with a reduction in creatinine at 24 h from 2.85 mg/dL to 1.5 mg/dL (*p* = 0.0002) and a reduction in urea from 192.8 mg/dL to 129.2 mg/dL for the non-treated group to the HYBRI group, respectively (*p* = 0.0335, Figure 3). Interestingly, the major difference between groups was observed on the second day after ischemia, when both parameters began to decrease in the HYBRI group but peaked in the non-treated group. At this point, the creatinine level of the non-treated group increased to 3.02 mg/dL compared to the reduction to 0.89 mg/dL (*p* = 0.001) for the group treated with the HYBRI protein. Furthermore, in the case of blood urea the slope on the second day continued increasing, with a significant difference from 281.3 mg/dL in the non-treated group to 76.4 mg/dL in the HYBRI group (*p* = 0.0022). There were no serum creatinine and urea level fluctuations in the sham group.

#### 2.2.2. Renal Histopathology and Immunohistochemistry

A significant reduction in histological damage was observed from a total inflammatory score of 6.75 in the non-treated group to 3.2 in the HYBRI group (*p* = 0.0021), while there was no inflammation in the sham group (Figure 4).

Despite no significant differences, the HYBRI group showed a reduction of almost half in CD3 infiltrating cells (from 13.2 in the non-treated group to 8 in the HYBRI group). An enrichment of FoxP3+ infiltrating cells (Tregs) in the kidney was observed (from 0.7 in the non-treated group to 1.3 in the HYBRI group, with a trend to significance (*p* value= 0.075)). Even more, creating a ratio between infiltrating FoxP3+ cells and infiltrating CD3+ cells, animals treated with HYBRI showed a percentage of 39.1% in contrast to 8.9% in non-treated animals (*p* value= 0.069). The treatment also revealed a significant reduction in CD68 infiltrating cells from 3.3 in the non-treated group to 1.8 in the HYBRI group (*p* < 0.0001).

#### 2.2.3. Peripheral Blood and Spleen Cell Populations

There were no significant differences between groups in a flow cytometry analysis in spleen cell subpopulations. Regarding the Peripheral blood mononuclear cell (PBMC) analysis, the only differences seen were in the percentage of FoxP3+ cells (Tregs) in the CD4+ cells. These cells were expanded from 5% to 13% the next day after ischemia in the HYBRI group. In contrast, in the non-treated group, the difference was lower, with an initial 4% to 9% twenty-four hours after ischemia. Similar trends were observed in the percentage of CD25+ cells in both the CD3+ and CD4+ cell subsets.

#### 2.2.4. Circulating Inflammatory Cyto/Chemokines

The serum levels of MCP1 and RANTES (also known as CCL5) at the end of the study decreased significantly with the HYBRI treatment. In addition, concerning the IP-10, IL-12, and MIP-1⍺ levels, treatment with HYBRI in ischemic kidneys significantly reduced those cytokines to values similar to those of sham rats. A partial but non-significant reduction in the serum IL-2 levels was also observed in the HYBRI group compared to the non-treated rats (Figure 5).

#### 2.2.5. Renal Gene Expression

After analyzing both the TaqMan low density array (TLDA) and Polymerase chain reaction (PCR) techniques with a CT method analysis, a few significant differences were observed comparing the HYBRI group with the non-treated group. In this regard, the CTLA4 (*p* = 0.0133) and FoxP3 (*p* = 0.0435) gene expression were significantly reduced in the non-treated group compared to the HYBRI group.

In HYBRI animals, there was a trend for lower expressions of Ccl7, Ccr5, Cxcl10, Fcnb, Ptprc, S100a6, Trl2, Tlr4, HGF, RORC, and VCAM genes compared to non-treated animals. Interestingly, the degree of reduction in the expression of those genes in the group treated with HYBRI led to values closer to those of the sham group, and, contrarily to non-treated rats, they were not statistically different, thus bringing the treated kidneys closer to a healthier genotype.

The Pathwax database results for all these genes showed that the HYBRI treatment affected nine molecular pathways related to cellular processes, such as apoptosis, necroptosis, and focal adhesion (see Appendix A). It also affected twelve environmental information processing-related pathways, such as cytokine–cytokine receptor interaction; cell adhesion molecules; and NF-kappa B, MAPK, JAK-STAT, and TNF signaling pathways as the most relevant. Of note, HYBRI also affected thirty-three pathways related to human diseases, with the PD-1 expression pathway as the most significant. Twenty-one organismal systems-related pathways such as NK cell-mediated cytotoxicity; Th1, Th2, and Th17 cell differentiation; the chemokine signaling pathway; T and B-cell and toll-like receptor signaling pathways were also affected. The gene expression results showed that almost all the analyzed genes are regulated in the IRI setting, but just thirteen of them are significantly addressed by the HYBRI treatment.

### 2.3. Rat Allogeneic Kidney Transplant Model

#### 2.3.1. Effects of HYBRI on Renal Function and Survival

HYBRI treatment resulted in an improvement in renal function in the allogeneic transplant model. At the early ischemic peak 24 h after kidney transplantation, despite no significant differences HYBRI treatment reduced the serum creatinine from 1.4 to 1.1 mg/dL and the serum urea levels from 185.8 to 157.2 mg/dL. Furthermore, HYBRI treatment showed a significant reduction in creatinine levels on day 7 after kidney transplantation (*p* = 0.0171), when kidney function begins to decline due to allogeneic kidney rejection (Figure 6).

There were also significant mortality differences between the studied groups. The initial number of twenty-one non-treated animals was reduced to seven at the 21st day, representing a 33% survival rate (Figure 7). Among the surviving animals, only one lived to the end of the study at day 90 (4.8%). The mean survival of this group was 23 days. On the other hand, the administration of the HYBRI protein during the firsts 6 days of the study significantly improved the animal survival up to 57 days, more than double that in the group without treatment. This data was significant in the Log rank Mantel Cox test of the Kaplan Meier curve. On the 21st day of the study, a survival rate of 80% of the group was seen (*p* = 0.021), and this finally decreased to 30% at the end of the study at day 90, a higher percentage than the almost 5% corresponding to the untreated group (*p* = 0.005). 

#### 2.3.2. Renal Histopathology and Immunohistochemistry

As seen in Figure 8, conventional kidney histology reflected a significant score reduction from 8.1 in the non-treated group to 3.5 in the HYBRI group (*p* < 0.001). Regarding kidney infiltrating cells, there were significant differences in CD3+, where the mean of 24.9 cells/hpf in the non-treated group was reduced to 10.3 cells/hpf in the HYBRI-treated group (*p* = 0.0237). Macrophage infiltration was also significantly reduced with HYBRI treatment, where the mean semi-quantitative score of 2.97 in the non-treated group was decreased to 1.94 with HYBRI treatment (*p* = 0.0015). Humoral effect was analyzed with C4d staining, and significant reductions were found with the HYBRI treatment in both studied areas (*p* = 0.0287 for glomeruli and *p* = 0.0034 for peritubular capillary).

#### 2.3.3. Circulating Inflammatory Cyto/Chemokines

A significant reduction of up to three-fold in IP-10, RANTES, and MCP1 was observed in rats treated with HYBRI compared with non-treated rats. Animals treated with HYBRI had reduced MIP1⍺ values, but they were not statistically different because there was a huge dispersal in non-treated rats. Finally, non-significant reduction in IFNγ and IL-12 was seen (Figure 9).

## 3. Discussion

Ischemia reperfusion damage is a deleterious response in acutely injured kidneys in several clinical situations. In this setting, costimulatory lymphocyte signals have been involved or modulated successfully [41,42,43,44,45]. We here show a positive modulation of both warm renal ischemia and allograft response by means of a dual and opposite costimulatory targeting, combining two coinhibitory receptors, CTLA4 and PDL2.

After the theoretical design of the construct and the genetic transient insertion in the CHO mammalian cell line, HYBRI was synthesized and its in vitro activity was assessed in the mixed lymphocyte reaction [46] with successful inhibition of T cell response. As a first validation, the well-known SPR technique showed the great affinity of our newly designed HYBRI protein with the CD80 and PD-1 targets, thus confirming that the binding sites of the proteins that make up the structure do not condition their binding avidity. In this regard, HYBRI protein presents its effective binding to both CD80 and PD-1 simultaneously, which indicates that HYBRI can block the CD28-CD80 costimulatory pathway while stimulating the PD-1-PD-L2 coinhibitory pathway. These bindings are effective despite the approximately 70% protein homology shared between species, given that HYBRI is a fully human protein [47,48]. Consequently, these results support the protein affinity to its designed targets, appearing as an interesting option in the inflammatory animal models addressed in the present study. Despite the reported similar KD values of the CD80 and HYBRI CTLA4 interaction, the KD values of the PD-1 and HYBRI’s PD-L2 interaction are higher than the values found in the literature [31,49,50,51,52]. Similarly to the increased affinity of the modified PD-L2 described by Philips et al. [51], PD-L2 molecules in the HYBRI construct may have suffered conformational changes, which might account for its increased binding to PD-1. Nevertheless, in both rat models HYBRI shows therapeutic efficacy, which suggests that the binding is high enough to exert biological effects.

The dual and opposite costimulatory targeting conferred by HYBRI have induced a protective effect against the kidney warm ischemia model, as the renal function and structure after the insult is greatly improved in the treated rats. This effect may be based on CD80 costimulatory blocking, which reduces the macrophage and T cell migration and, also, may promote T cell apoptosis [19,53,54,55,56]. The PD-1/PD-L2 pathway may have also produced Treg clonal expansion and diminished the T effector cell expression [24,34,57]. These pathways appear to be well addressed by the protein, although the expressions of these molecules and their ligands are not exclusive to the APC and T cells and, in some cases, can be bidirectional [34]. For example, renal tubular epithelial cells (RTECs), podocytes, and other renal cells also express CD80 and PD-L2 on their surface [58,59,60]. This may suggest that the desired homeostatic effect is not specific to APCs but can also be exerted on other semi-professional cells in antigen presentation. As ischemia is present in both groups, it can be seen that the expansion of Tregs promoted by the administration of HYBRI plays an essential role in homeostasis. Thus, the observed reduction in macrophages and T cells in the kidney together with the increase in regulatory T cells indicates the cellular modulation by the HYBRI treatment of post ischemic renal inflammatory damage. In addition, the significant decline in circulating inflammatory-related mediators reflects the efficacy of HYBRI in modulating the inflammatory response at the systemic level [[61],[62],[63],[64],].

A daily peripheral cytometric study showed only changes between study groups for Tregs expansion on the first day after ischemia. The results suggest that the CD25+ population experienced clonal expansion early after ischemia injury, also including the FoxP3+ subset, but that the Tregs subset is additionally expanded with the HYBRI treatment. At the end of the study, peripheral and spleen population assessment did not reveal significant cell differences between the groups despite a significant reduction in renal inflammatory infiltrate being found with HYBRI treatment. Thus, peripheral blood or spleen compartments may not reflect organ parenchymal changes where there is a real inflammatory environment with myriads of attractive and activation signals. The HYBRI treatment modulated the kidney inflammatory status towards a more attenuated inflammatory response and the enrichment of the Tregs population. In the allogeneic setting with a stronger immune response, the effect of HYBRI treatment may have offered two benefits. On the one hand, the reduction in the inflammation derived from cold ischemia reperfusion damage, but also an intense modulation of the adaptive response of T and possibly B cells given the reduction in C4d deposition.

A gene array analysis showed differences between the sham group and ischemia-reperfusion groups, but few significant differences between the non-treated and HYBRI groups. Thus, the regulation of some of the studied genes can be observed. These genes are related to both the inflammation and expression of cell surface proteins such as CD80 and CD86. In this regard, the HYBRI treatment may produce a reduction in CD80 and CD86 cell surface expression, an important point in this immunomodulation strategy [19,53,54]. It is also remarkable that the protein HYBRI modifies only some of the pathways related to IRI, despite showing a large and significant therapeutic effect in vivo.

Regarding early cold ischemia injury in the allogeneic transplant model, a slight amelioration in renal functional parameters is observed in the group treated with HYBRI. In this complex surgical model, it is hard to discern ischemic damage among all surgical attritions, contrarily to the easier native warm ischemia. In this second proposed model, however, the effect of the HYBRI protein plays an essential role in kidney rejection, where the treated group improved the renal function on days 7 and 9 after transplantation and prolonged the survival of animals to more than double that of non-treated rats. Recently, it has been shown that there is early PD-1 expression in renal allograft dynamics. Blocking this PD-1/PD-L1 negative costimulatory pathway during the first week doubled the number of PD-1-expressing CD8 and CD4 cells, causing terminal acute rejection [29]. These findings with an approach completely opposite to our PD-1, which activates this negative pathway, suggest the essential role of this pathway in modulating T cell expansion and cytokine production in the allograft setting. Similar studies with PD-L1 and PD-L2 blocking antibodies in a model of IRI also aggravated the renal damage, again suggesting the protecting role of this coinhibitory pathway [24].

To summarize, the SPR study has shown the exquisite binding affinity of HYBRI to both targets and the advantage of binding simultaneously in vitro. Regarding in vivo studies, the HYBRI protein may effectively mitigate damage from warm ischemia reperfusion. In the context of the kidney transplant model, the HYBRI protein also prevents kidney rejection even by treating animals only during the first week. Therefore, this dual and opposite protein effect on costimulatory pathways can promote several immunomodulatory mechanisms, leading to protecting the kidney from multiple immunoinflammatory states.

## 4. Materials and Methods

### 4.1. HYBRI Characterization and Binding Affinity Assessment with Surface Plasmon Resonance (SPR) Technique

Surface plasmon resonance (SPR) analyses were performed at 25 °C on a Biacore T200 (GE Healthcare BioSciences AB, Uppsala, Sweden). The HYBRI construct was diluted in acetate buffer pH 5 (50 µg/mL and immobilized on a CM5 chip (GE) by amine coupling in the active flow channels (Fc) at two different ligand densities. Briefly, the surface was activated with a mixture of 1:1 mixture of 0.1 M NHS (N-hydroxysuccinimide) and 0.1 M EDC (3-(N,N-dimethylamino) propyl-N-ethylcarbodiimide) at a flow rate of 5 μL/min, followed by the injection of ligand or buffer for the active or reference channel, respectively. Ethanolamine solution was injected in both channels in order to block the remaining reactive groups of the surface. Analytes (recombinant murine PD-1 and CD80, R&D systems) were prepared in PBS-P in serial dilutions (0, 15.6, 62.5, 125, 250, 500, 1000, and 2000 nM) and were injected in parallel in two active and reference channels at 30 µL/min flow for 90 s and a dissociation time of 240 s. Blanks were included for double referencing. Experiments were also conducted to assess the effect of PD-1 presence on the CD80 binding to Hybri. For this, concentrations series of CD80 (0–2000 nM) in the presence of PD-1 (250 nM) were included. Kinetic and affinity constants were calculated using the Biacore T200 evaluation software 2.0 after reference and blank subtraction, and sensorgrams were fitted according to the 1:1 Langmuir model.

### 4.2. Animals and Surgical Procedures

All the procedures were performed following the Guidelines of the European Committee on Care and Use of Laboratory Animals and Good Laboratory Practice. Rats from Charles River, Spain, were housed with 12 h dark/light cycles and at a constant temperature. The animals were fed *ad libitum* with standard diet and water. For both ischemia and transplant surgical models, a combined anesthesia based on ketamine (75 mg/kg), atropine (0.05 mg/kg), and valium (5 mg/kg) was used. A single intramuscular injection of ciprofloxacin (5 mg) was administered after the surgery.

### 4.3. Renal Warm Ischemia Model

In this model, Lewis rats received intraperitoneally one single dose of 20 mg/kg of HYBRI administered 24 h before bilateral renal ischemia of 40 min (*n* = 10, Hybri). Non-treated animals were used as control group (*n* = 10, Vehicle), and they received one single intraperitoneal administration of 300 µL of PBS 24 h before the bilateral renal ischemia of 40 min. The sham group received neither treatment nor ischemia and only laparotomy as a technique control group was performed (*n* = 3, Sham).

Blood samples were obtained from the tail vein at time zero and on daily basis until sacrifice on day 7 after ischemia. Animals were euthanized under anesthesia on the seventh day, blood was obtained by aortic puncture, and the spleen and kidneys were processed.

### 4.4. Allogeneic Transplant Model (Cold Ischemia)

In a model of renal allotransplantation, binephrectomized Lewis rats received a single kidney transplant from a Wistar donor rats. HYBRI group (*n* = 10) animals were treated with administration of 500 µg of HYBRI intraperitoneally 2 h before transplantation and the six following days. Non-treated (*n* = 21) animals were used as a control group with the administration of 200 µL of PBS with the same schedule.

Blood samples of both groups were obtained from the tail vein at time zero and on days 1, 3, 5, 7, 9, 11, and 14 and weekly after the transplant. Surviving animals at day 90 were euthanized under anesthesia, blood was obtained by aortic puncture, and the spleen and kidneys were processed.

### 4.5. Renal Function

Renal function was analyzed daily on warm ischemia model and on days 0, 1, 3, 5, 7, 9, 11, and 14 and weekly on the allogeneic transplant model. Serum creatinine and urea measurements were performed following Jaffe’s and GLDH reactions (Olympus Autoanalyzer AU400, Hamburg, Germany) in the Veterinary Clinical Biochemistry Laboratory of Universitat Autonoma de Barcelona.

### 4.6. Histological and Immunohistochemistry Studies

Coronal kidney slices 1–2 mm thick were fixed in buffered 4% of formalin, paraffined, and stained with H&E. Kidney sections from the ischemia-reperfusion study were evaluated by a blinded pathologist, examining the tubular necrosis, dilation, interstitial edema, and cellular infiltrate. Abnormalities were scored on semiquantitative scale from 0 to 4 as follows: 0, no abnormalities; 1, changes <25%; 2, changes 25–50%; 3, changes 50–75%; and 4, changes >75%. A mean group score was attained from all the individual parameters.

For the allogeneic study, sections were analyzed by a blinded pathologist for tubulitis, interstitial infiltration, vasculitis, glomerulitis, tubular necrosis and glomerular necrosis following the Banff criteria for acute/active lesions scoring, held at the Ninth Banff Conference on Allograft Pathology in La-Coruna, Spain, on 26 June 2007.

For immunohistochemistry techniques, paraffin tissue sections were stained for CD3 (Abcam, Cambridge, UK), C4d (Hycult Biotech, Uden, Netherlands), CD68 (AbD Serotec, Raleigh, NC, USA), and FoxP3 (Novus Biologicals, CO, USA). Sections were immune peroxidase labeled and revealed by diaminobenzidine (Sigma, Madrid, Spain). For the CD3, CD68, and FoxP3 measurement, at least 10 hpf were taken to analyze and count. For the C4d measurement, a semiquantitative scale from 0 to 3 was employed. Negative controls from immunostained-matched sections without primary antibodies were used.

### 4.7. Peripheral Blood and Spleen Cell Subsets Characterization by Flow Cytometry

Daily peripheral blood from warm ischemia model and blood before transplant and on days 7 and 21 for the transplant model was collected in heparin tubes and separated in cytometry tubes to add the antibodies and lysis buffer needed to continue with the cytometry protocol.

Splenocytes were preserved at −80 °C after its isolation using the Ficoll (GE Healthcare) density gradient. Standard methods were used to thaw, wash, and recover the cells. An incubation of 25 min in the dark at room temperature with different monoclonal antibodies was performed for the cytometry technique using FACS Canto II Cytometer and the subsequent analysis using FACS DIVA software (BD Biosciences, San Jose, CA, USA). The antibodies were titrated, mixed, and formulated for optimal staining performance.

Cocktail T/B/NK (BD 558495) containing anti-CD3 APC, anti-CD45RA FITC, and anti-CD161 PE for T, B, and NK cell detection; anti-CD43 PE (Biolegend 202812), anti-CD161 AF647 (AbD Serotec MCA1427A647), and anti-ED9 FITC (AbD Serotec MCA620F) for monocytes detection; and anti-CD3 PerCP efluor (eBioscience 46-0030-80), anti-CD4 FITC (eBioscience 11-0040-85), anti-CD25 APC (eBioscience 17-0390-82), and anti-Fox P3 PE (eBioscience 12-4774-42) for Tregs detection. Cell membrane and nucleus permeabilization was needed for Tregs cytometric detection.

### 4.8. Measurement of Serum Levels of Inflammatory Factors by Luminex Fluorescent Assay

The determination of IFNγ, IL-2, IL-12/IL-23p40, MIP-1⍺ (CCL3), RANTES (CCL5), IP-10 (CXCL10), and MCP-1 (CCL2) concentrations in serum was conducted using Luminex ProcartaPlex™ Multiplex Immunoassay (Thermo Fisher Scientific, Waltham, MA, USA) following the manufacturer’s instructions. Results were calculated from the calibration curves and expressed in pg/mL.

### 4.9. Quantification of Gene Expression in Kidneys from Warm Ischemia Model

Snap-frozen rat kidneys from the ischemia-reperfusion study were stored at—80 °C. RNA samples with an A260/280 ratio of ≥1.8 were extracted using the PureLinkTM RNA MiniKit (Invitrogen, Madrid, Spain) following the manufacturer’s instructions. For the reverse transcription, High-Capacity cDNA reverse Transcription Kit (Applied Biosystems, Madrid, Spain) was used following the manufacturer’s instructions.

The tissue expression of immune-inflammatory mediators was quantified by Taqman Low Density Array microfluidic cards (ABI-PrismH-7700, Applied Biosystems, Madrid, Spain) using the comparative CT method: Emp1/Emp3/Lgals1/Lgals3/Reln/S100a6/S100a8/S100a9/Socs3/Socs5/Tnfrsf12a/Fcnb/CD4/CD40/CD80/CD86/Tnf/Nfkb1/Ctla4/Tlr2/Tlr4/Tlr6/Ccl2/Ccl3/Ccl5/Ccl7/Cxcr3/Cxcl10/Cxcl11/Ccr2/Ccr3/Ccr5/Ptprc/Junb/Igsf6/Ido2/Il2/Il9/Il10/Il15/Il17a/Tfcp2/Mapk9/Spp1/Bcl6 and eukaryotic 18S as an endogenous reference.

Moreover, the expression of genes validated in recent studies from our group [1] were quantified by TaqMan real-time PCR (Applied Biosystems, Madrid, Spain) using the comparative CT method: C1qa/C1qc/C1r/C1s/CD19/CD276/CD44/Cxcl3/Cxcr4/Fas/FoxP3/Hgf/Ifng/IL-10ra/IL-11/RORC/Socs1/Sox9/TLR8/TLR9/Tnfrsf1b/Vcam and GAPDH as endogenous reference. Controls, which were composed of distilled water, were negative for the target and reference genes.

A pathway analysis of single gene sets, it was set up using the online PathwaX.sbc.su.se web server, which applies the BinoX algorithm to KEGG pathways and FunCoup networks [65].

### 4.10. Statistical Analysis

For a statistical analysis, the Student’s t-test compared two conditions, whereas the ANOVA was employed for the comparison of multiple conditions. Repeated measures ANOVAs were used to analyze differences in various parameters due to HYBRI treatment throughout the follow-up. Nonparametric analysis was used as needed. For the comparative CT method, the value shown is the one obtained by performing the Wilcoxon rank sum test when comparing the data of the study treatment with the data of the control treatment. The Kaplan-Meier method was performed and analyzed using the Mantel Cox test for a comparison of the survival distributions in the two groups. A value of *p* < 0.05 was considered significant. Data are given as a mean ± s.e.m.

## Figures and Tables

**Figure 1 ijms-22-01216-f001:**
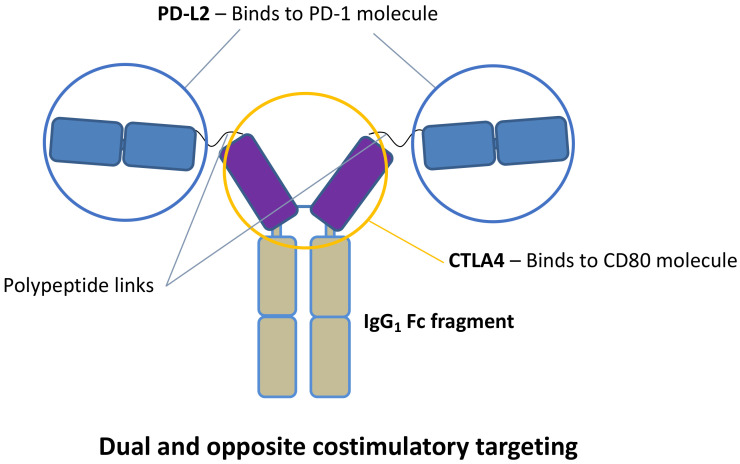
HYBRI protein structure.

**Figure 2 ijms-22-01216-f002:**
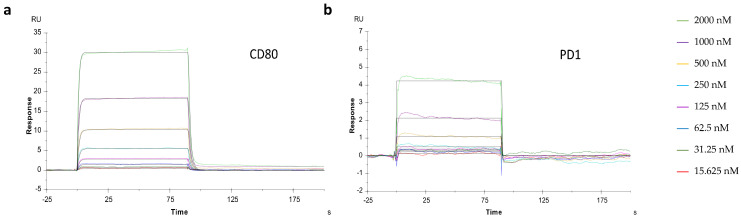
Representative sensorgrams of CD80 (**a**) and PD1 (**b**) binding to HYBRI. HYBRI was bound to CM5 chip and CD80 and PD1 serial dilutions (15.625, 31.25, 62.5, 125, 250, 500, 1000, and 2000 nM) represented in distinct colors were injected for 90 s. Experimental data are shown as color traces and black traces represent the fitted data according to the 1:1 Langmuir binding model.

**Figure 3 ijms-22-01216-f003:**
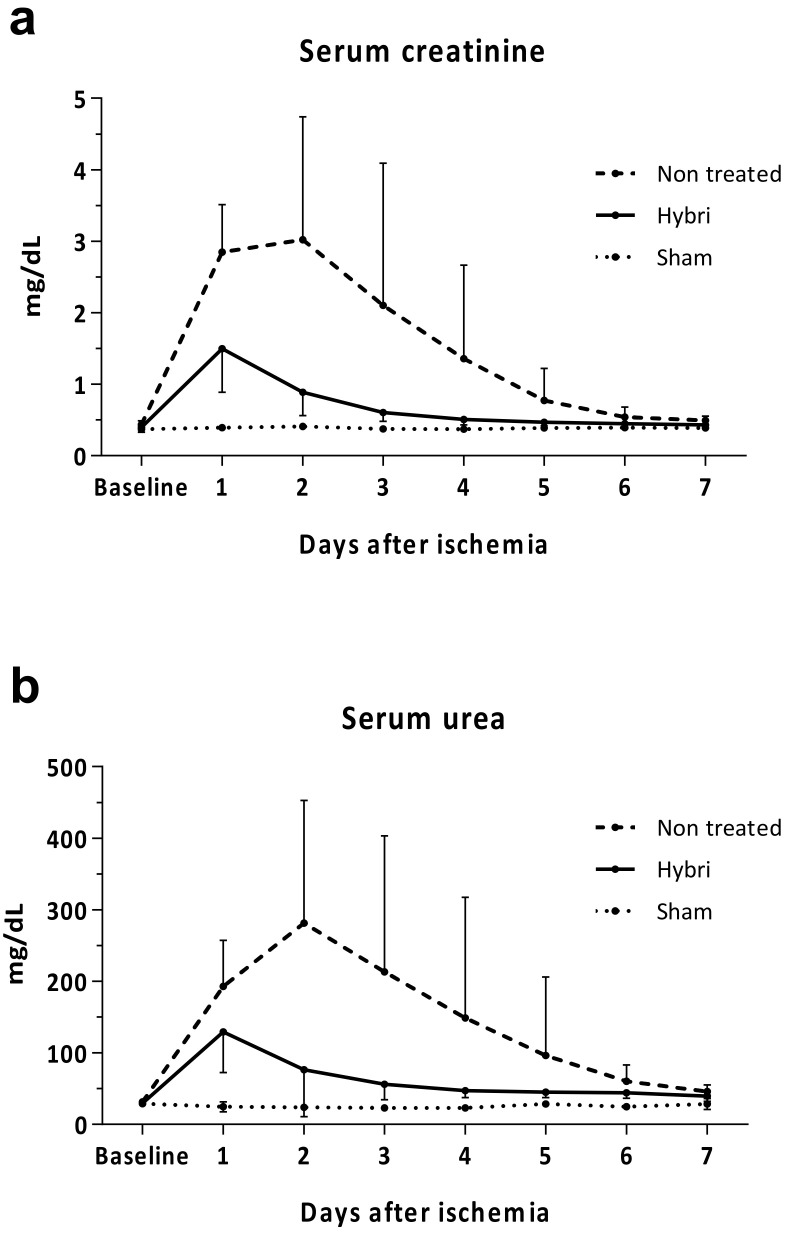
Functional renal parameters throughout the ischemia-reperfusion model for all treatments, including serum creatinine (**a**) and serum urea (**b**). Values are expressed in mg/dL. Data are expressed as mean.

**Figure 4 ijms-22-01216-f004:**
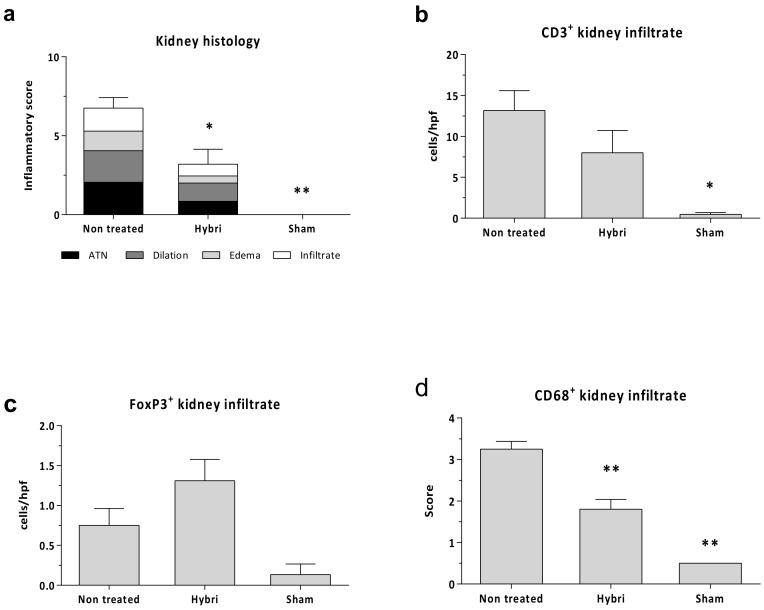
Renal histopathology parameters for ischemia-reperfusion injury in the three studied groups. Semi-quantitative inflammatory score values (**a**) for Hematoxylin-Eosin stain. Cells per hpf values for renal immunofluorescence CD3+ (**b**) and FoxP3+ (**c**) kidney infiltrate, and semi-quantitative score for CD68+ surface intensity (**d**). Data are expressed as a mean ± SEM. *, *p* < 0.05 vs. non-treated; **, *p* < 0.01 vs. non-treated.

**Figure 5 ijms-22-01216-f005:**
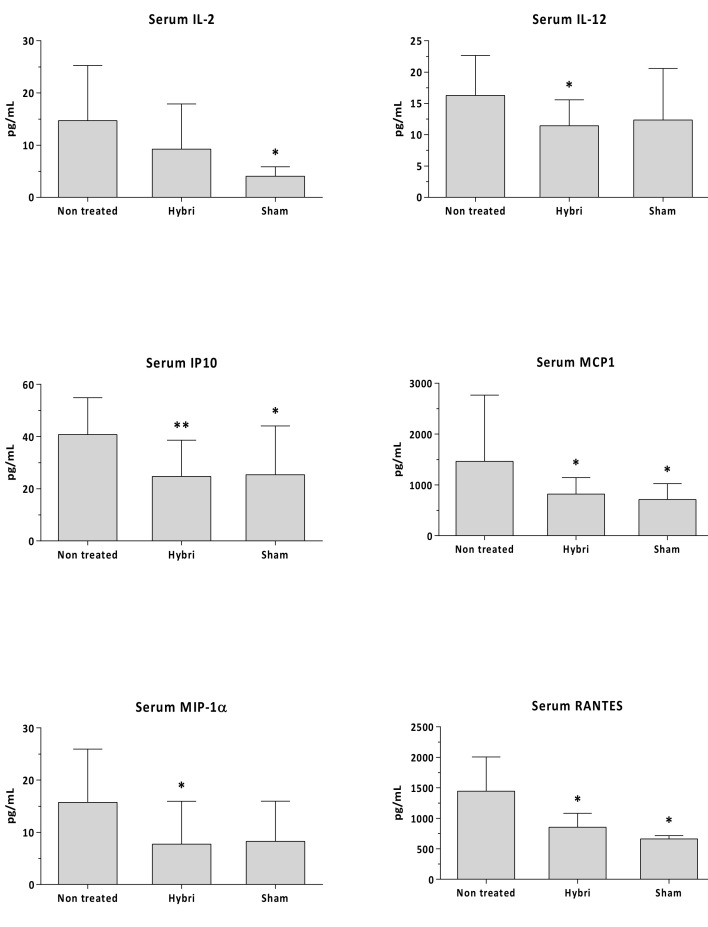
Circulating IL-2, IL-12, IP10, MCP1, MIP-1α, and RANTES in renal warm ischemia model measured by Luminex assay. Data are expressed as a mean ± SEM. *, *p* < 0.05 vs. non-treated; **, *p* < 0.01 vs. non-treated.

**Figure 6 ijms-22-01216-f006:**
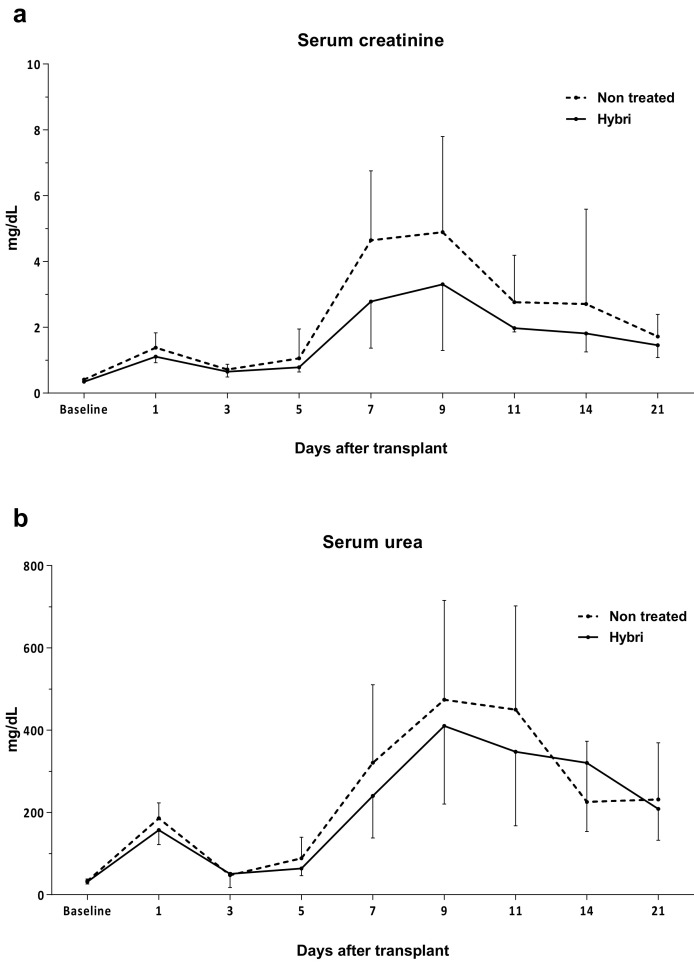
Functional renal parameters throughout the allogeneic transplant model for HYBRI treatment and non-treated groups. (**a**) Serum creatinine and (**b**) serum urea levels for the initial 21 days of the study. Values are expressed in mg/dL. Data are expressed as means.

**Figure 7 ijms-22-01216-f007:**
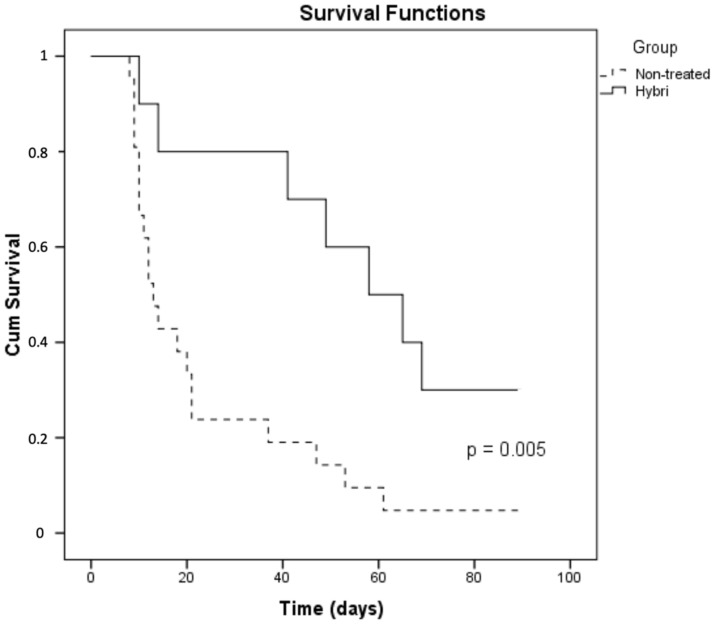
Survival differences with the Kaplan Meier analysis of the two studied groups in the rat allogeneic transplant model.

**Figure 8 ijms-22-01216-f008:**
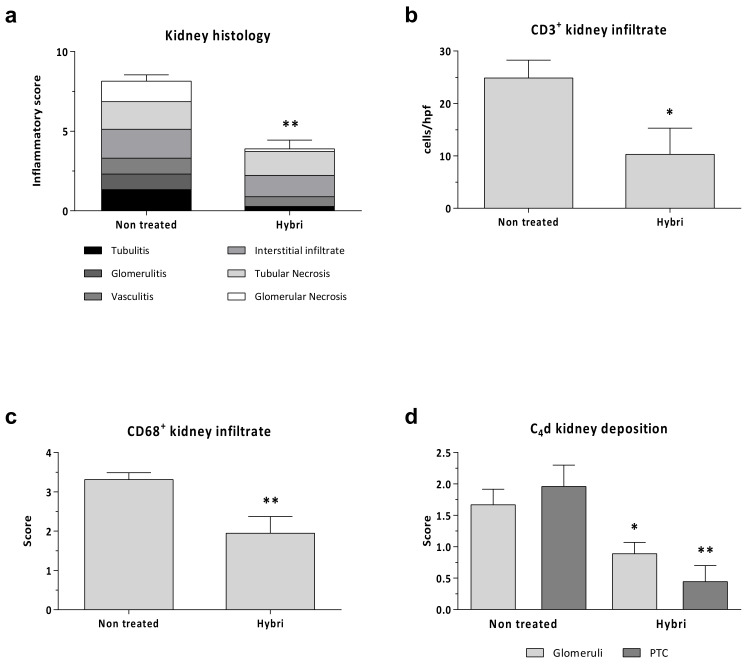
Renal histopathology parameters for allogeneic transplant model in the two studied groups. Semi-quantitative inflammatory score values (**a**) for Hematoxylin-Eosin stain. Cells per hpf values for renal immunofluorescence CD3+ (**b**) and semi-quantitative score for CD68+ surface intensity (**c**) and C4d kidney deposition (**d**). Data are expressed as a mean ± SEM. *, *p* < 0.05 vs. Non-treated; **, *p* < 0.01 vs. Non-treated.

**Figure 9 ijms-22-01216-f009:**
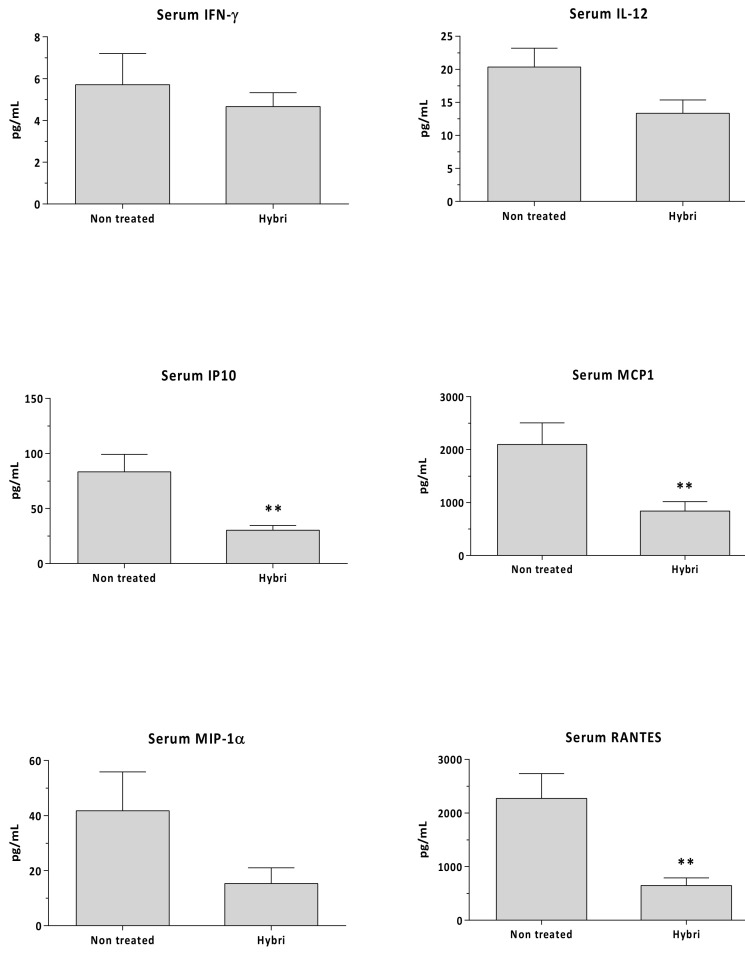
Circulating IFNγ, IL-12, IP10, MCP1, MIP-1α, and RANTES in allogeneic transplant model measured by Luminex assay. Data are expressed as a mean ± SEM. **, *p* < 0.01 vs. non-treated.

**Table 1 ijms-22-01216-t001:** CD80 and PD1 bind to HYBRI. Affinity constants (KD) were calculated from SPR analysis in the steady state and from the kinetics analysis.

	Steady State	Kinetics		
		ka (1/Ms)	kd(1/s)	KD (M)
CD80	3.72 × 10^−6^	1.57 × 10^5^	5.68 × 10^−1^	3.62 × 10^−6^
PD-1	5.29 × 10^−5^	6.37 × 10^4^	3.19	4.91 × 10^−5^

## Data Availability

The data presented in this study are available on request from the corresponding author.

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
