# Peer review of "Dual and Opposite Costimulatory Targeting with a Novel Human Fusion Recombinant Protein Effectively Prevents Renal Warm Ischemia Reperfusion Injury and Allograft Rejection in Murine Models"

_ijms, 2021, doi:10.3390/ijms22031216_

Round 1
Reviewer 1 Report
This is a very interesting paper reporting a novel approach to ameliorate ischemia-reperfusion injury, also in the setting of allo-transplantation, in a rat kidney model. The paper is well written and very convincing.
Comment:
1. Error bars should be added in Figure 6.
Author Response
Point 1: Error bars should be added in Figure 6.
Response 1: Thank you for your suggestion. We have added the Error bars in Figure 6 and also in figure 3.
Reviewer 2 Report
In this experimental study, Guiteras and colleagues evaluated a newly-developed hybrid protein (Hybri) with a postulated effect on CTLA4 and PD-L2 ligands and therefore potentially acting on several pathways involved in renal ischemia reperfusion injury (IRI) and kidney allograft rejection.
Main objectives of their research were to demonstrate that Hybri binds to the target molecules using Surface Plasmon Resonance technique, to assess Hybri effects on an animal model of renal IRI, and to evaluate Hybri effects on an animal model of kidney transplantation.
According to results, Hybri was able to bind both CD80 and PD1 with a dose-dependent direct interaction. High affinity as well as rapid association and dissociation for both the targets were also demonstrated. The rat model of IRI showed that a single prophylactic administration of the experimental compound was associated with better renal function (creatinine and urea concentrations), less histological damage (total inflammatory score), lower number of infiltrating monocytes, lower serum levels of MCP1, RANTES, IP-10, IL-12, MIP-1α, and increased renal CTLA4 and FoxP3 gene expression than non-treated controls. Furthermore, beneficial effects were observed in the allogenic kidney transplant model. In fact, Hybri treatment (once daily before and after surgery) resulted in higher animal survival rate, improved allograft function, better Banff acute/active lesions scoring, lower number of infiltrating CD3+ cells and monocytes, reduced renal C4d deposition, and lower levels of serum IP-10, RANTES, and MCP1 than non-treated rats.
The manuscript is well written, interesting, and novel. The experimental models are properly constructed and the study design is appropriate. Results are overall convincing and the conclusions appear adequately supported. Considering the detrimental impact of delayed graft function and drug-related toxicity on renal recipient and transplant survivals, the development of new, effective, and potentially safer immunosuppressive medications is vital.
Major comments:
1) In the introduction, the authors should better explain the relationship between IRI and delayed graft function (DGF) as well as the clinical impact of DGF on transplant-related outcomes (increased risk of rejection, inferior allograft function, difficult management of immunosuppression in the early post-transplant phase, and biopsy-related complications).
2) In the introduction, the authors should briefly describe available evidence supporting the use of co-stimulation blockage (i.e., belatacept) in clinical kidney transplant setting (T-cell-mediated rejection, anti-HLA antibody production, antibody-mediated rejection, safety profile compared to calcineurin inhibitors).
3) In the discussion, the authors say that “The delayed nephrectomy at the seventh day after ischemia might be a limitation. Shortening the study follow-up could be an alternate option to avoid the re-establishment of many signalling pathways”. Have the authors considered removing one kidney in the early IRI phase and the other one at a later stage? Assuming similar baseline characteristics for both the kidneys, this option could have provided more information.
Minor comments:
1) The title of the manuscript is somehow misleading as it does not suggest that the study was performed using an animal model.
2) In the introduction, I would recommend to cite the study from Shim Y et al. (Kidney Int 2020;98(4):897-905) on PD-1 and PD-L1 in murine kidney transplantation.
3) Results section 2.2.3. (Peripheral blood and spleen cell populations) is quite difficult to follow and it would benefit from a more detailed description of the findings. The sentence “these results suggest that CD25+ etc.” sounds more like an explanation and as such should be part of the discussion.
4) In the methods section, it would be interesting to know what was the rationale behind the use of different Hybri doses before IRI and before transplant.
5) Being supported by results obtained in animal models, the conclusions seem too optimistic. I would encourage to change “Hybri effectively prevents damage from warm ischemia-reperfusion” with “Hibri may effectively mitigates damage from warm ischemia-reperfusion”. Similarly, I would change the sentence “Hybri has enough potency to prevent rejection”.
6) Figure 6 (b) is redundant
Author Response
Comments and Suggestions for Authors
In this experimental study, Guiteras and colleagues evaluated a newly-developed hybrid protein (Hybri) with a postulated effect on CTLA4 and PD-L2 ligands and therefore potentially acting on several pathways involved in renal ischemia reperfusion injury (IRI) and kidney allograft rejection.
Main objectives of their research were to demonstrate that Hybri binds to the target molecules using Surface Plasmon Resonance technique, to assess Hybri effects on an animal model of renal IRI, and to evaluate Hybri effects on an animal model of kidney transplantation.
According to results, Hybri was able to bind both CD80 and PD1 with a dose-dependent direct interaction. High affinity as well as rapid association and dissociation for both the targets were also demonstrated. The rat model of IRI showed that a single prophylactic administration of the experimental compound was associated with better renal function (creatinine and urea concentrations), less histological damage (total inflammatory score), lower number of infiltrating monocytes, lower serum levels of MCP1, RANTES, IP-10, IL-12, MIP-1α, and increased renal CTLA4 and FoxP3 gene expression than non-treated controls. Furthermore, beneficial effects were observed in the allogenic kidney transplant model. In fact, Hybri treatment (once daily before and after surgery) resulted in higher animal survival rate, improved allograft function, better Banff acute/active lesions scoring, lower number of infiltrating CD3+ cells and monocytes, reduced renal C4d deposition, and lower levels of serum IP-10, RANTES, and MCP1 than non-treated rats.
The manuscript is well written, interesting, and novel. The experimental models are properly constructed, and the study design is appropriate. Results are overall convincing, and the conclusions appear adequately supported. Considering the detrimental impact of delayed graft function and drug-related toxicity on renal recipient and transplant survivals, the development of new, effective, and potentially safer immunosuppressive medications is vital.
Major comments:
Point 1: In the introduction, the authors should better explain the relationship between IRI and delayed graft function (DGF) as well as the clinical impact of DGF on transplant-related outcomes (increased risk of rejection, inferior allograft function, difficult management of immunosuppression in the early post-transplant phase, and biopsy-related complications).
Response 1: We have modified the paragraph in accordance with the reviewer’s recommendations:
Delayed graft function (DGF) is the clinical manifestation of ischemia-reperfusion injury (IRI) in human renal transplantation. The clinical impact of DGF on transplant-related outcomes is an increase in the risk of rejection, inferior allograft function, difficult management of immunosuppression in the early post-transplant phase, and the need for repeated renal biopsies with the related associated complications [1]– [3].
Point 2: In the introduction, the authors should briefly describe available evidence supporting the use of co-stimulation blockage (i.e., belatacept) in clinical kidney transplant setting (T-cell-mediated rejection, anti-HLA antibody production, antibody-mediated rejection, safety profile compared to calcineurin inhibitors).
Response 2: We added additional information for a better understanding:
The costimulatory blocker belatacept (a mutated version of CTLA4Ig) was approved for immunosuppression in renal transplantation in the early 2000s. The observed advantages of belatacept over cyclosporine include better graft function, preservation of renal structure and improved cardiovascular risk profile [21]. Concerns associated with belatacept are a higher frequency of cellular rejection episodes, but protocol biopsies at 1 year showed a lower incidence of chronic allograft nephropathy with belatacept compared to cyclosporine [22]. Accordingly to Parsons et al, Belatacept can also reduce HLA class I antibodies in a significant proportion of highly sensitized recipients [23].
Point 3: In the discussion, the authors say that “The delayed nephrectomy at the seventh day after ischemia might be a limitation. Shortening the study follow-up could be an alternate option to avoid the re-establishment of many signalling pathways”. Have the authors considered removing one kidney in the early IRI phase and the other one at a later stage? Assuming similar baseline characteristics for both the kidneys, this option could have provided more information.
Response 3: We decided to delete the sentence according to your suggestions. We agree that this sentence is merely speculative and does not introduce any critical information.
Regarding the second suggestion, our main interest in the study was to evaluate the clinical evolution of renal function and some kidney data at the seventh day using Hybri in the warm renal ischemia model. A proof of concept with our new protein. Obtaining kidney samples in early stages is out of the interest of present study. This may be the subject of future studies.
Minor comments:
Point 4: The title of the manuscript is somehow misleading as it does not suggest that the study was performed using an animal model.
Response 4: We have added to the title “in murine models”
Point 5: In the introduction, I would recommend to cite the study from Shim Y et al. (Kidney Int 2020;98(4):897-905) on PD-1 and PD-L1 in murine kidney transplantation.
Response 5: We have added this paper to better understand the role of PD-1 cell signaling involvement in the mechanisms of acute rejection in organ transplantation.
Early PD-1 expression has been demonstrated in renal allograft, which was considered essential to modulate T-cell expansion and cytokine production. Blocking the PD-1/PD-L1 coinhibitory pathway during the first week after transplantation doubled the number of PD-1–expressing CD8 and CD4 cells infiltrating the graft, mainly in the interstitium [29].
Point 6: Results section 2.2.3. (Peripheral blood and spleen cell populations) is quite difficult to follow and it would benefit from a more detailed description of the findings. The sentence “these results suggest that CD25+ etc.” sounds more like an explanation and as such should be part of the discussion.
Response 6: Result section 2.2.3. has been reformulated to better understanding. Additionally, we moved the suggested sentence to the discussion.
Point 7: In the methods section, it would be interesting to know what was the rationale behind the use of different Hybri doses before IRI and before transplant.
Response 7: In the transplant model, we followed a similar therapeutic schedule previously reported on the use of CTLA4-Ig in experimental transplantation: Perico et al. Kidney Int. 1995; 47: 241.
Point 8: Being supported by results obtained in animal models, the conclusions seem too optimistic. I would encourage to change “Hybri effectively prevents damage from warm ischemia-reperfusion” with “Hibri may effectively mitigates damage from warm ischemia-reperfusion”. Similarly, I would change the sentence “Hybri has enough potency to prevent rejection”.
Response 8: We fully agree that we were enthusiastic in our conclusions, following your suggestions we have softened the message.
Point 9: Figure 6 (b) is redundant
Response9: Completely agree.
Reviewer 3 Report
The subject of the study is interesting, and the researchers did a lot of work to evaluate the role of a new synthetic protein capable of binding CD80 and PD1 in kidney I-R injury and allograft rejection. Certainly, such a rigorous study deserves publication providing the authors to improve some points and note some limitations.
- The authors should re-check the manuscript to correct some inaccuracies. For instance, in the abstract, the CTLA4 pathway is not a co-stimulatory pathway but an inhibitory pathway. The CD28 pathway, which shares the same ligands with CTLA4, is co-stimulatory. Also, the term hybri protein in the title is too general and does not provide information about the study's subject. There are many such points throughout the manuscript.
- The “hybri” protein, except for CD80, should also be able to engage the CD86, the other co-stimulatory protein on the surface of antigen-presenting cells. This, was not evaluated and should be commented on.
- CD80 is on the surface of antigen-presenting cells, and according to the literature, renal tubular epithelial cells also express CD80 (Kidney International 1999; 56:. 1551–1559). CD28 and CTLA4 are on the surface of T-cells. PDL2 on the surface of renal cells, and PD1 is on the surface of T-cells. Is there any possibility of the “hybri” to cross-link T-cells with APCs and RTECs? And what would be the possible consequences? A discussion would be well-come. Also, CD80 is expressed on podocytes as well (J Am Soc Nephrol. 2016; 27: 963–965).…
- A comparison with anti-CD80 therapy (such as the belatacept) alone, and anti-PD1 therapy alone (such as nivolumab) alone was not performed. This is a limitation that should be noted.
- Continuing the above, it is known that anti-PD1 therapy alone induces allograft rejection in kidney transplant recipients since it potentiates T-cell reactivity. How does the anti-PD1 component of “hybri” not do the same? Is there any possibility for the “hybri” to activated instead of inhibiting the PD1 signal transduction? A discussion and possible further experiments are required.
- The manuscript needs English editing.
Author Response
The subject of the study is interesting, and the researchers did a lot of work to evaluate the role of a new synthetic protein capable of binding CD80 and PD1 in kidney I-R injury and allograft rejection. Certainly, such a rigorous study deserves publication providing the authors to improve some points and note some limitations.
Point 1: The authors should re-check the manuscript to correct some inaccuracies. For instance, in the abstract, the CTLA4 pathway is not a co-stimulatory pathway but an inhibitory pathway. The CD28 pathway, which shares the same ligands with CTLA4, is co-stimulatory. Also, the term hybri protein in the title is too general and does not provide information about the study's subject. There are many such points throughout the manuscript.
Response 1: We have modified the abstract sentence where we talk about the costimulatory and coinhibitory pathways for a better understanding of the concepts. We have also modified some sentences in the introduction and discussion according to your suggestions. We agree with the reviewer that the terms costimulatory and coinhibitory can lead to confusion because these pathways can be activated or inhibited depending on their own nature or the ligand used (agonist or blocker). We have done our best to give the manuscript a better understanding.
We have deleted “hybrid” and replaced by “human fusion recombinant protein”. Perhaps we have not explained it correctly enough. The term HYBRI corresponds to the name we have given to our human fusion recombinant protein. To improve the understanding, we have changed Hybri for HYBRI.
We understand that the information about the study subject is not given by “HYBRI”, but by the “dual and opposite costimulatory targeting”. What we did not use are the terms CTLA4 and PD-L2 in the title to simplify it. Readers will find in the main body text of the manuscript the complete explanation of the study subjects.
Point 2: The “hybri” protein, except for CD80, should also be able to engage the CD86, the other co-stimulatory protein on the surface of antigen-presenting cells. This, was not evaluated and should be commented on.
Response 2: It is planned to evaluate the CTLA4 – CD86 interaction using SPR technique. We firstly decided to check the CTLA4 binding to CD80 because it seems to be the predominant ligand according to the literature, and CD80 is broadly expressed on activated APCs (Jansson et al. J Immunol 2005).
Point 3: CD80 is on the surface of antigen-presenting cells, and according to the literature, renal tubular epithelial cells also express CD80 (Kidney Int 1999; 56:. 1551–1559). CD28 and CTLA4 are on the surface of T-cells. PDL2 on the surface of renal cells, and PD1 is on the surface of T-cells. Is there any possibility of the “hybri” to cross-link T-cells with APCs and RTECs? And what would be the possible consequences? A discussion would be well-come. Also, CD80 is expressed on podocytes as well (J Am Soc Nephrol. 2016; 27: 963–965).…
Response 3: According to your comments, RTEC and podocytes express CD80 which can bind to CD28 activating this costimulatory pathway or CTLA4 inhibiting it. Also, CTLA4 from Hybri can bind to these CD80 and then inhibit this pathway. Therefore, the protein may exert the same action in any receptor or four different cell types (APCs, T cells, RTECs and podocytes), conferring the expected immunomodulation. Similarly occurs with PD-L2.
Regarding the “cross-link” possibility, if we have understood correctly, this is the chance of the protein to interfere at the same time in two different cell synapsis among these cell types. Hybri protein presents CTLA4 which can bind to CD80 in one cell, and also presents two PD-L2 which can bind to PD-1 in the same cell or in a second or a third cell. However, we assume that one molecule of Hybri only acts between just two cell types because of its size, length, and structure. The nature of the cell to which it binds, will be anyone of the cell types described previously. The polypeptide links that attach CTLA4 and PD-L2 in the Hybri arms are conformed by a few amino acids. However, this is just an assumption and we are now performing structural studies such as crystallographic analysis that will give more information.
We have added this rational in the discussion:
These pathways appear to be well addressed by the protein, although the expression of these molecules and their ligands are not exclusive to the APC and T cells and in some cases can be bidirectional [34]. For example, RTECs, podocytes or other renal cells also express CD80 and PD-L2 on their surface [58]–[60]. This may suggest that the desired homeostatic effect is not specific to APCs but that it can also be exerted on other semi-professional cells in antigen presentation.
Point 4: A comparison with anti-CD80 therapy (such as the belatacept) alone, and anti-PD1 therapy alone (such as nivolumab) alone was not performed. This is a limitation that should be noted.
Response 4: The main objective of the study was to test the immunomodulatory effect of the protein and to demonstrate its therapeutic effect using these two rat models. The comparison with the domains alone should add further information, which now is a limitation of our study. But in this first stage of the Hybri assessment, we sought a PoC and finding the proper dosage. Further studies will assess groups of animals using Belatacept and PD-L2. We will internally discuss the suitability of introducing groups with Nivolumab or other check-point inhibitors.
Point 5: Continuing the above, it is known that anti-PD1 therapy alone induces allograft rejection in kidney transplant recipients since it potentiates T-cell reactivity. How does the anti-PD1 component of “hybri” not do the same? Is there any possibility for the “hybri” to activated instead of inhibiting the PD1 signal transduction? A discussion and possible further experiments are required.
Response 5: Hybri molecule is designed with PD-L2 domain to induce immunosuppression along with CTLA4. We assume that the binding of this PD-L2 with the cellular PD-1 exerts inhibitory response on T cell dynamics, acting as an agonist not as a blocker. Recent studies have shown an overexpression of T cell PD-1 on the first seven days after transplantation, which was overcome by the administration of anti-PD-1 (Shim).
In kidney IRI, administration of PD-L1 or PD-L2 blocking Abs prior to mild or moderate kidney IRI significantly exacerbated the loss of renal function, renal inflammation, and acute tubular necrosis compared with mice receiving isotype control Abs. Interestingly, blockade of both PD-1 ligands resulted in worse injury, dysfunction, and inflammation than did blocking either ligand alone. Genetic deficiency of either PD-1 ligand also exacerbated kidney dysfunction and acute tubular necrosis after subthreshold ischemia [24].
We have added in the discussion:
Recently, it has been shown that there is an early PD-1 expression in renal allograft dynamics. Blocking this PD-1/PD-L1 negative costimulatory pathway during the first week doubled the number of PD-1–expressing CD8 and CD4 cells causing terminal acute rejection [29]. These findings with an approach completely opposite to our PD-1 pathway activation argue the essential role of this pathway to modulate T-cell expansion and cytokine production in the allograft setting. Similar studies with PD-L1 and PD-L2 blocking antibodies in a model of IRI also aggravated the renal damage, again arguing the protecting role of this coinhibitory pathway [24].
Point 6: The manuscript needs English editing.
Response 6: Ok. We have re-read the manuscript several times improving the overall writing.
Round 2
Reviewer 3 Report
The authors addressed all issues adequaltely.